# Assessment of K-Struvite Precipitation as a Means of Nutrient Recovery from Source Separated Human Urine

Işık Kabdaşlı [1,*], Sezen Kuşçuoğlu [1], Olcay Tünay [1] and Alessio Siciliano [2]

[1] Environmental Engineering Department, Civil Engineering Faculty, İstanbul Technical University, Ayazağa Campus, Istanbul 34469, Turkey; sezenk@gmail.com (S.K.); tunayol@itu.edu.tr (O.T.)

[2] Laboratory of Environmental Sanitary Engineering, Department of Environmental Engineering, University of Calabria, 87036 Rende, Italy; alessio.siciliano@unical.it

\* Correspondence: kabdasli@itu.edu.tr; Tel.: +90-212-2856-586

**Abstract:** The impact of nutrients on the environment, particularly on water bodies, has led to extensive studies for nutrient control. Within this context, studies have been focused on source separation of human urine from domestic wastewater to recover nutrients. Potassium is one of the most important components of human urine. However, data on potassium removal or recovery are quite limited except for some indirect information through use of zeolites for mostly ammonia removal. Potassium struvite or K-struvite ($MgKPO_4 \cdot 6H_2O$) is a sparingly soluble salt belonging to the struvite family that has the potential of being used as a means of potassium and phosphate recovery from segregated human urine. This study aimed to assess the potential of K-struvite precipitation for control and recovery of nutrients. Within this context, K-struvite precipitation experiments were performed on both synthetically prepared samples and synthetic human urine solution to determine effect of operation parameters i.e., pH, stoichiometry, and temperature on potassium recovery performance. Results indicated that process performance as well as type of solid phases co-precipitated with K-struvite were closely related to initial potassium concentration, pH, and reaction stoichiometry. At pH 10, the potassium recovery efficiency was maximized up to 87% by application of 100% excess dose of Mg and P for both synthetic samples and synthetic human urine solution. On the other hand, application of excess dose of K did not provide any improvement in K recovery efficiency. The effect of temperature on solubility of K-struvite was insignificant at the temperature range of 24–90 °C. Solid phase analyses confirmed that K-struvite was co-precipitated with either $Mg_3(PO_4)_2$, $MgNaPO_4 \cdot 7H_2O$, or $MgHPO_4 \cdot 7H_2O$ depending on pH and stoichiometry instead of a pure compound.

**Keywords:** human urine; K-struvite precipitation; nutrient recovery; operation parameters; solid phases

## 1. Introduction

Magnesium ammonium phosphate ($MgNH_4PO_4 \cdot 6H_2O$; MAP), a member of the struvite family, has been used with success for nitrogen removal and as a means of nutrient recovery. MAP precipitation has a wide area of application such as source separated urine [1–4], anaerobic treatment discharge [5], and a number of industrial wastewaters, including leather tanning [6–10], slaughterhouse [11], animal husbandries [12], textile printing [13], landfill leachate [14–16], and membrane process concentrates [17]. This has led scientists to the other recoverable nutrient, potassium, which is also abundant in source-separated urine. The struvite family of compounds also includes magnesium potassium phosphate (K-struvite), a sparingly soluble salt, which has been considered as a potential method for potassium removal and recovery. Potassium is also a valuable nutrient and can be recovered from the source separated urine [18–20]. MAP precipitation is relatively easy to conduct due to two reasons. The first reason is its low solubility with respect to K-struvite and the second is the two major co-precipitating solids ($Mg(OH)_2$

and $Mg_3(PO_4)_2$) can be avoided at the optimum pH of around 9.5 of MAP [21]. Therefore MAP precipitation results in an almost pure solid of magnesium ammonium phosphate. K-struvite precipitation suffers from both higher solubility and unavoidable co-precipitates which make the process a less efficient one with respect to MAP precipitation. Theoretical assessment of the K-struvite precipitation is not easy, and determination of thermodynamic constants of the process is still on the way. Struvite type of compounds has the general formula of $M^{2+}N^+xO_4 \cdot nH_2O$, where $M^{2+}$: Mg or Ca; $N^+$: K, Rb, Cs, Tl, or Na, $NH_4^+$; $x$: P or As, and $n = 6$–8. Additional information about these compounds can be found elsewhere [22–24]. K-struvite ($MgKPO_4 \cdot 6H_2O$; MPP) is a member of this family. MPP is also a sparingly soluble salt, but its dissolution is not congruent. It cannot be precipitated as a pure compound, but rather as a mixture of other sparingly soluble solids in urine and other sodium containing solutions. For the case of source separated urine co-precipitating solids are MPP and magnesium sodium phosphate ($MgNaPO_4 \cdot 7H_2O$; MSP). The exact solubilities of both co-precipitates (MPP and MSP) are not known. For MPP, pKsp values are given as 10.6 [25], 11.7 [26], and 12.2 ± 0.253 [27]. These figures exhibit a wide range. For MSP, Xu and co-workers [27] calculated the pKsp as 11.6 ± 0.253 which is within the range of pKsp values of MPP indicating close Ksp values for MPP and MSP. Thus, they inevitably precipitate together. As the pKsp of MAP is 13.26 [28,29], the existence of ammonia inhibits the MPP precipitation. Therefore, studies in the literature were conducted on synthetically prepared urine that does not contain ammonia or real urine pre-treated to remove ammonia [24,27,30,31]. Warmadewanthi and Lui [32] claimed that magnesium phosphates (Bobierrite, $Mg_3(PO_4)_2 \cdot 8H_2O$, pKsp = 25.2, and tri magnesium phosphate $Mg_3(PO_4)_2 \cdot 22H_2O$, pKsp = 23.1) were also co-precipitated with MPP. The pH values over which these solid phases begin to precipitate were reported as 12 and 9 for bobierrite and tri magnesium phosphate, respectively. Xu et al. [24] pointed out that $Mg_3(PO_4)_2 \cdot 8H_2O$ may precipitate after pH 10 and 11 depending on P:K ratio in MAP precipitation. A similar evaluation was made by Lee et al. [21] and Zhang et al. [33] for MAP precipitation. On the other hand, Taylor et al. [25] reported experimental results of MPP precipitation conducted with a pH between 10.42–10.87 where both bobierrite and tri magnesium phosphate co-precipitated. However, no other study of MPP precipitation cited in this paper has shown any evidence of magnesium phosphate precipitation, including crystal identification means such as X-ray diffraction (XRD). Thermodynamic models have not included these compounds as well.

So far MPP precipitations have been conducted mostly in source separated urine and synthetic urine solution, except for a study using 1/5 diluted urine in a draft tube and baffle reactor [30]. Usage of synthetic solutions has the advantage of being free of impurities and salts that are present in urine and the experimental results on these solutions may be more representative. The present study attempts to evaluate K-struvite precipitation using a wide range of initial potassium concentration (10–250 mM) in batch systems with synthetic solutions. The effects of pH, reaction stoichiometry, time, and temperature on K recovery performance were investigated. The applicability of K-struvite precipitation to the synthetic human urine was explored in terms of process performance. Solid phase analyses were also realized.

## 2. Materials and Methods

### 2.1. Samples

Synthetic samples were prepared using $KH_2PO_4$ and $MgCl_2 \cdot 6H_2O$, and KOH, $NaH_2PO_4$, and $MgCl_2 \cdot 6H_2O$ for K-struvite precipitation experiments with stoichiometric and excess doses, respectively. While synthetic human urine was prepared, urea was excluded from the original recipe given elsewhere [27,30,34]. The recipe used in the present study is given in Table S1 (see Supplementary Material). All reagents were of analytical grade and purchased from Sigma–Aldrich Chemicals (Sigma–Aldrich Corporation, Wilwaukee, WI, USA).

## 2.2. Analytical Procedure and Instruments

Synthetic samples were prepared using $CO_2$-free deionized water as explained in our previous study [35]. All analyses were accomplished as defined in Standard Methods for the Examination of Water and Wastewater [36]. pH was measured with Orion 920A model pH meter (Thermo Fisher Scientific, Beverly, MA, USA). Jenway PFP7/C Research Flame Photometer (Analytik Jena GmbH, Jena, Germany) was used for potassium measurement. Solid phase analyses were made using X-ray diffractometer (XRD; Rigaku Dmax 2200, Rigaku Corporation, Tokyo, Japan) and scanning electron microscopy (SEM; Philips XL30 SFEG, FEI/Philips, Hillsboro, OR, USA). The deionized water with conductivity of 0.055 $\mu S\, cm^{-1}$ was produced by Sartorius Arium 611-UV Water Purification System (Sartorius, Göttingen, Germany).

## 2.3. Precipitation Experiments

Erlenmeyer flasks with stoppers containing 500 mL equipped with magnetic stirrers to ensure homogenous mixing were used as precipitation vessels. After reagent addition to $CO_2$-free deionized water, if required, initial pH values were adjusted under flash-mixing conditions using a NaOH solution (0.1–1 N). During the course of slow-mixing, solution pH was measured once a day and adjusted when necessary. At the end of MPP precipitation, all samples were filtrated through Sartorius 0.45 mm membrane filters before analysis.

## 2.4. Equilibrium Time and Temperature

In order to determine the duration (time) needed to reach the equilibrium, the 7-day precipitation and a long duration of one-month equilibrium time results were compared for 250 mM initial concentration at pH 9.0 (Table S2, see Supplementary Material). The differences between the two durations were slight, indicating that the 7-day equilibrium duration was considered adequate for experimental evaluations. This finding is consistent with the result of Xu et al. [27]. After one week storage, maximum relative tolerances were measured as 3.2% and 5.2% for P and K, respectively, in their experiments. Based on these data, the reaction time required to reach equilibrium was selected as 60 min.

The effect of temperature on solubility of K-struvite and potentially precipitating other solids was also explored through a MPP precipitation conducted at pH 9.0 by employing a 333 mM initial dosage of a stoichiometric K/Mg/P solution. This experiment was realized at room temperature (24 °C), 30 °C, and 90 °C. The results are displayed in Table S3 (see Supplementary Material). As seen from the table, solubilities or supernatant concentrations after precipitation were almost identical, indicating that the effect of temperature was negligible in the range studied. The temperature dependency of magnesium ammonium phosphate (magnesium struvite) has been assessed in the literature and similar results have been found [29]. Based on these data all precipitation experiments were performed in a temperature-controlled laboratory section at 25 ± 0.1 °C for a week ensuring equilibrium condition.

## 3. Results and Discussion

### 3.1. Effect of pH

pH is one of the key parameters in precipitation processes which involve acid-base reactions. The first experimental evaluation was made using a 250 mM initial concentration with a K/Mg/P molar ratio of 1/1/1 to ensure high level of saturation for a wide pH range of 6 to 11. The results of MPP precipitation performed using $KH_2PO_4$ and $MgCl_2 \cdot 6H_2O$ as precipitating agents are presented in Figure 1 (Table S4).

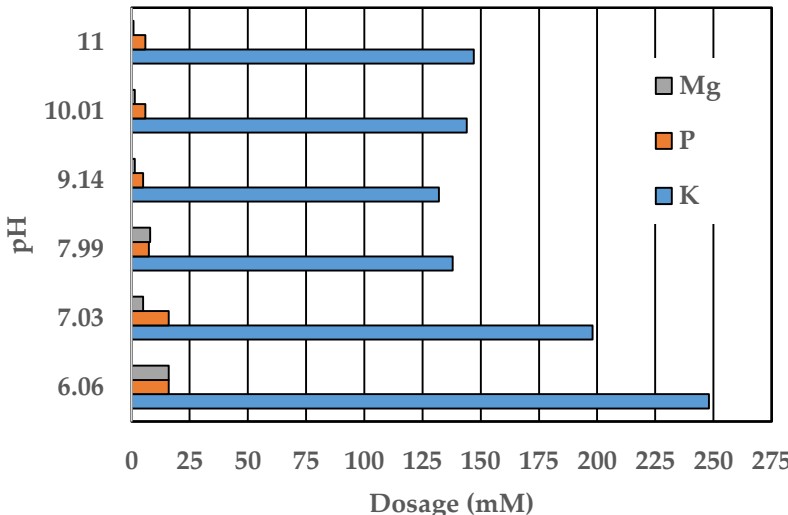

**Figure 1.** Effect of pH on K-struvite precipitation (K/Mg/P: 250/250/250 mM).

The results indicated that K concentration together with P and Mg decreased in all experiments with pH values between 7.03 and 11.00. At pH 6.06, however, K concentration remained unchanged, while P and Mg concentration significantly reduced from 250 mM to 16 mM. Considering that the formation of MSP was impossible since no pH adjustment was made using NaOH during this run, the significant reduction of both Mg and P might be attributed to the precipitation of $MgHPO_4$. At pH 7.03, K removal with precipitation is evident and only K salt precipitating is MPP. The precipitate was assumed to be a mixture of $MgHPO_4$ and MPP because K removal was 20% and NaOH addition for pH adjustment was extremely low. Results of precipitation at pH 7.99 could be interpreted in a similar manner as that of 7.03, while MPP precipitation becoming significantly higher as K removal was 44%. The precipitation conducted at pH 9.14 yielded slightly higher K removal than that of pH 7.99, indicating almost the same extent of MPP precipitation. $MgHPO_4$ and $Mg(OH)_2$ were not likely to precipitate leaving only a possibility of MSP. Results of pH 10.01 and pH 11.00 were quite similar with one another and seemed to have a similar mechanism with that of pH 9.14, with a small reduction in K recovery efficiency. Although residual K concentrations differed, the mechanism depicted above was in accordance with the experimental results of Taylor et al. [25]. Xu et al. [24] found in their experimental study with synthetic urine that K removal efficiencies decreased after pH 12, while pH 10 and 11 yielded efficiencies very close to each other and represented optimum pH for K removal. They explained the K removal efficiency drop at pH 12 with formation of $Mg(OH)_2$ and $Mg_3(PO_4)_2$. Huang et al. [31] tested low grade MgO as a means of K removal from urine that was pre-treated to remove ammonia. Their experimental data indicated that K precipitation remained practically unchanged after pH of approximately 10.7. Therefore, the general picture obtained from the experiments seems consistent with theoretical as well as experimental results given in the literature.

In the light of the above evaluations, forthcoming MPP precipitation experiments were run within the pH range of 8–10. In the present paper, we also intentionally avoided to use the term "optimum pH" since it does not seem meaningful for a system with more than one solid phase precipitating. Instead, the term "maximum removal or recovery" was used, which is relative to system composition and aim of the work.

To assess the effect of pH on K recovery performance, another set of precipitation experiments was run at pH 9 and 10 with an initial concentration of 100 mM equimolar Mg, K, and P, which represented the upper concentration of K in human urine. The results are outlined in Table 1. As seen from the table, while residual P and Mg concentrations were similar with those of the experiments with 250 mM initial concentration, resulting K concentrations were lower, and K recovery efficiency was also reduced (at pH 9 dropped

from 47% to 25%). Although the initial concentration of potassium used was high, being near 4000 mg/L, the potassium removal efficiencies were still low and less than 30%.

**Table 1.** Effect of pH on K-struvite recovery (K/Mg/P: 100/100/100 mM).

| pH | Initial (mM) | | | Final (mM) | | |
|---|---|---|---|---|---|---|
| | K | P | Mg | K | P | Mg |
| 9.03 | 100 | 100 | 100 | 75 | 1.3 | 1.3 |
| 10.05 | 100 | 100 | 100 | 72 | 0.2 | 6 |

Figure 2 displays the results of precipitation experiments initiated at 10 mM stoichiometric dose. As seen in the figure, K recovery efficiencies had a maximum of 7% and did not change by pH. Data given in this subsection (Table S5), altogether indicated that K recovery efficiencies constantly decreased as initial K concentration reduced and became practically inapplicable for 10 mM (390 mgK $L^{-1}$).

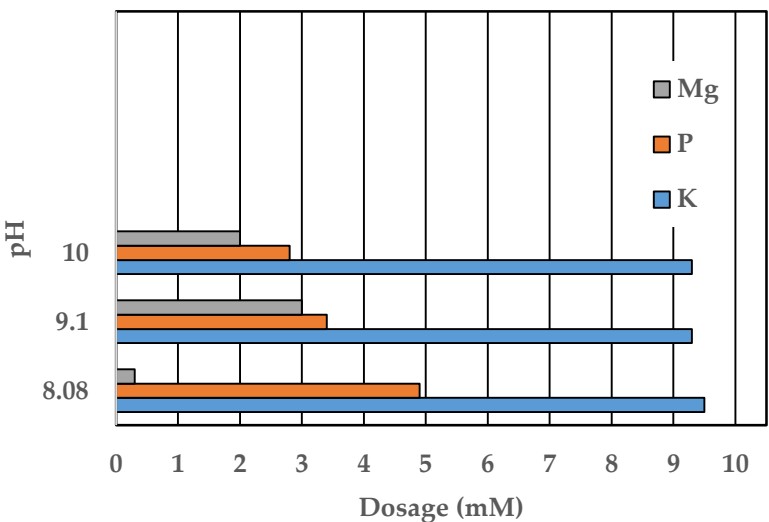

**Figure 2.** Effect of pH on K-struvite recovery (K/Mg/P: 10/10/10 mM).

The overall results indicated that the precipitation process at stoichiometric doses may only be used for partial recovery, with about 40% potassium removal and pH values from 8 to 10.

### 3.2. Precipitation with Different Stoichiometry

To investigate the effect of the reaction stoichiometry on the process performance, MPP precipitation experiments with over-stoichiometric dose were planned and realized. KOH, NaH$_2$PO$_4$, and MgCl$_2$·6H$_2$O were utilized as reagents in these experiments. In the first step, K was dosed over stoichiometric to Mg and P at the excess K ratios 20%, 34% and 44%. The results are outlined in Table 2.

**Table 2.** Effect of excess K dose on process performance.

| pH | Initial (mM) | | | Final (mM) | | |
|---|---|---|---|---|---|---|
| | K | P | Mg | K | P | Mg |
| 9.02 | 180 | 150 | 150 | 121 | 1.65 | 1.6 |
| 9.02 | 180 | 125 | 125 | 112 | 2.4 | 1.3 |
| 8.99 | 250 | 187 | 187 | 190 | 2.2 | 0.9 |

As seen in Table 2, for 125 and 150 mM doses of Mg and P, application of 180 mM K resulted in very low residual P and Mg levels. K removals were limited to 32% and 37% for

150 mM and 125 mM doses, respectively, exhibiting much less than KMgPO$_4$ stoichiometry. For a K dose of 250 mM, P and Mg levels were again low, but K removal was at 24% and the residual amount was 190 mM, whereas it was 132 mM for 250 mM stoichiometric (K/Mg/P: 1/1/1) dosing at pH 9.14 (Figure 1; Table S4). These results indicate that overdose of K was ineffective in controlling and improving the K recovery performance. These results may not mean much as far as the application of potassium recovery is concerned, but from the standpoint of theory it may indicate that the excess potassium cannot change the amount of co-precipitating solids.

In the second step, the effect of overdoses of Mg and P on K recovery efficiency was explored at pH values of 9 and 10. Table 3 shows the results of these experiments. For 180 mM K concentration, increasing both Mg and P doses from 187.5 to 250 mM improved K recovery efficiency from 24% to 50%. The improvement in K recovery efficiency was also evident at 100% overdoses of P and Mg for a wide range of initial K concentration from 6.25 to 125 mM. The maximum K recovery was obtained as 85% with 100 mM K concentration at pH 10.02. Comparison of K recovery efficiencies obtained at pH 9 with those of pH 10 showed that using pH 10 seemed more effective to enhance K recovery. Furthermore, in all experiments, residual Mg and P values were very low indicating simultaneous precipitation of magnesium phosphate solids. Therefore, over stoichiometric Mg and P application would help K recovery without an increase in residual Mg and P. This point is of importance for the application in that the added phosphate, which is another valuable nutrient is also recovered in the solids and is not lost in the effluent. In general Mg and P overdosing seems to be the only way to increase the K removal for the precipitation process to reach applicable limits.

**Table 3.** Results of precipitation experiments at overdoses of Mg and P.

| pH | Initial (mM) | | | Final (mM) | | |
|---|---|---|---|---|---|---|
| | K | P | Mg | K | P | Mg |
| 9.09 | 180 | 187.5 | 187.5 | 135 | 1.9 | 1.4 |
| 9.07 | 180 | 250 | 250 | 90 | 0.4 | 4 |
| 9.07 | 125 | 250 | 250 | 41 | 0.9 | 3 |
| 9.00 | 50 | 100 | 100 | 17 | 1.9 | 2 |
| 10.03 | 125 | 250 | 250 | 38 | 3 | 0.6 |
| 10.02 | 100 | 200 | 200 | 15 | 0.2 | 1.5 |
| 10.02 | 50 | 100 | 100 | 11 | 0.2 | 1.4 |
| 10.11 | 25 | 50 | 50 | 7 | 0.8 | 1.3 |
| 10.08 | 12.5 | 25 | 25 | 6 | 1 | 1.2 |
| 10.12 | 6.25 | 12.5 | 12.5 | 5 | 1.9 | 1.2 |

### 3.3. Dissolution of Precipitate

To determine the components and their composition of the precipitate produced in the experiments, a selected solid phase (obtained from the experiment run at 250 mM stoichiometric doses of K/Mg/P and pH 9.0) was completely dissolved in acid. The amount of solid dissolved was 8 gL$^{-1}$. The composition of the obtained solution was found as shown in Table 4.

**Table 4.** Composition of precipitate.

| | K | PO$_4$ | Mg | Na |
|---|---|---|---|---|
| mM | 6.5 | 19 | 19 | 7 |

The results clearly showed that amount of P and Mg were several folds greater than K indicating high amount of magnesium phosphate contributed to the solid. Existence of sodium was almost equal the amount of K, suggesting the presence of a sodium containing solid which was likely to be MSP [37]. The obtained solid phase stoichiometry is

Mg/K/Na/P: 2.9/1/1.01/2.9 and the stoichiometry of both MSP and MPP is 2/1/1/2. This finding reveals the fact that the solid phase comprised a mixture of MSP, MPP, and some magnesium phosphates. Therefore, even if the reliable solubility constants were available, a theoretical solution of the system and prediction of potassium removals would not be easy.

### 3.4. Potassium Recovery from Human Urine

Human urine is the principal source for potassium recovery using K-struvite precipitation. Therefore, we also conducted precipitation experiments on synthetic human urine (SHU) defined in the Materials and Methods section. Considering the findings in the relevant literature [24,27,30,38] that ammonia removal by a suitable process, such as stripping, prior to K-struvite precipitation is an essential step so as to enhance K recovery from SHU solutions, urea was not added to the SHU solution to avoid MAP precipitation. Thus, the main reactive species of SHU were K, Mg, and P. The results obtained from our experimental study and their interpretation strongly suggested that in such a system the solid phase would be composed of a mixture of different substances. Therefore, different stoichiometric doses were employed in these precipitation experiments. The obtained results are given in Table 5.

**Table 5.** Results of K-struvite precipitation for SHU solution.

| pH | K:Mg:P | Initial (mM) | | | Final (mM) | |
|---|---|---|---|---|---|---|
| | | K | P | Mg | K | P |
| 9.40 | 3/1/0.5 | 31.82 | 10.32 | 5.16 | 23 | 3.44 |
| 9.42 | 3/1/1 | 31.82 | 10.32 | 10.32 | 24 | 0.99 |
| 9.39 | 3/1/1.3 | 31.82 | 10.32 | 13.41 | 24 | 0.36 |
| 9.44 | 3/1/1.5 | 31.82 | 10.32 | 15.48 | 25 | 0.14 |
| 9.48 | 1/1/1 | 31.82 | 31.82 | 31.82 | 25 | 0.01 |
| 9.44 | 1/2/2 | 31.82 | 63.64 | 63.64 | 11 | 9.67 |
| 10.04 | 1/2/2 | 31.82 | 63.64 | 63.64 | 4 | 4.78 |

As seen in Table 5, the best result, through the experiments conducted using SHU solution with initial K concentration of 31.82 mM, was obtained as 87% K recovery with K/Mg/P ratio of 1/2/2 and at pH 10.04. With the same stoichiometry, but at pH 9.44, K recovery (67%) as well as P removal decreased. These may be related to lesser solubility of MPP at pH 10.4. These K recovery efficiencies are comparable with data published in the literature. For example, ammonium, K, and P removals were reported as 73%, 76%, and 68%, respectively, in the study Xu et al. [30] performed on ammonia stripped-diluted (1/5) real human urine in a laboratory-scale draft tube and baffle reactor. In another study, conducted in a pilot-scale fluidized bed reactor using SHU (recipe: 30.9 mM $KH_2PO_4$, 21 mM KCl, 0.7 mM $CaCl_2 \cdot 2H_2O$, 78.7 mM NaCl, 16.2 mM $Na_2SO_4$, and 2.86 mM $NH_4Cl$), for optimized operation conditions of pH of 10.5, Mg:P molar ratio of 1:1, and super saturation ratio of 3.0, K and P removal efficiencies were reported as $20-35\%$ and $80-90\%$, respectively [38]. This K removal efficiency is the same level with that of the precipitation experiment conducted at pH 9.48 with the K/Mg/P stoichiometry of 1/1/1. The condition of K/Mg/P ratio of 1/2/2 and pH 10.04, in a similar manner with that of synthetic solution experiment, provided a satisfactory result for K recovery.

### 3.5. Solid Phase Analyses

Further studies were conducted to assess the solid phases that were formed through the K-struvite precipitation. Within this context, three selected solid phases were analysed using x-ray diffraction (XRD) method (Figures S1–S3, see Supplementary Material). The first solid phase was the one obtained from the precipitation at pH 10.05 with stoichiometric dose (K/Mg/P: 1/1/1) for an initial K concentration of 100 mM. The result indicated the existence of magnesium phosphate and consistent with the discussion of this experiment as well as

with the experimental evaluation given by Taylor et al. [25]. Other solid phases analysed using XRD were the precipitates that resulted from the experiments at the pH 10 with the applications of over-stoichiometric (K/Mg/P: 125/250/250 mM) and stoichiometric (250/250/250 mM) Mg and P doses. Examination of these results showed the existence of sodium in the solid rather than magnesium phosphate, implying the precipitation of MSP as pointed out in the relevant literature [24,27,30].

Scanning electron microscope (SEM) photographs of solid phases obtained through the experiments were used to evaluate the morphology of the precipitates (Figures 3–5).

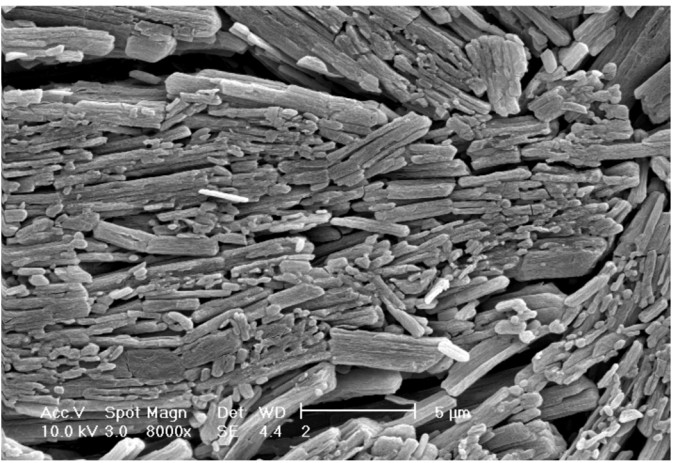

**Figure 3.** SEM analysis of solid products obtained from precipitation experiment performed at pH 9.14 and K/Mg/P: 250/250/250 mM.

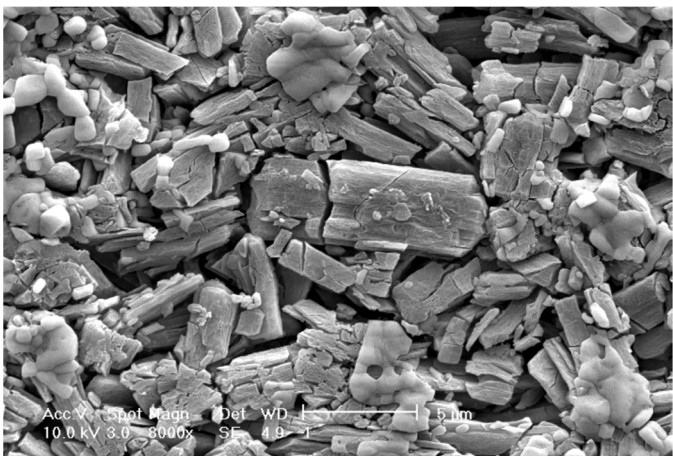

**Figure 4.** SEM analysis of solid products obtained from precipitation experiment performed at pH 10.05 and K/Mg/P: 100/100/100 mM.

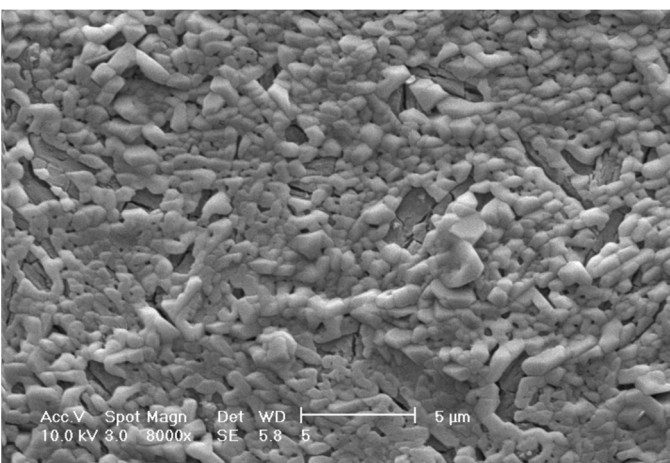

**Figure 5.** SEM analysis of solid products obtained from precipitation experiment performed at pH 10.03 and K/Mg/P: 125/250/250 mM.

In Figure 3, K-struvite crystals can be clearly seen as a similar structure with that of MAP having long needle like shape. In Figure 4, magnesium phosphate crystals with their spherical shape together with K-struvite can be observed. Their size was irregular. In the case of overdose of Mg and P, a mixture of crystals each having different structure was noticed (Figure 5).

## 4. Conclusions

The following conclusions could be drawn from the present study.

- In the case of synthetic samples, K recovery efficiency was found to be dependent on initial K concentration, reaction stoichiometry, and pH.
- K recovery efficiencies reduced as the initial K concentration decreased. The lowest K recovery efficiency (7%) was obtained for initial K concentration of 390 mg K $L^{-1}$ at stoichiometric dose.
- A 100% excess dose of Mg and P (K/Mg/P: 100/200/200 mM) yielded K recovery efficiency of 85% corresponding to a maximum one for synthetic samples at pH 10.02. Almost equal K recovery efficiency (87%) was obtained for synthetic human urine at pH 10.04 and 100% excess dose of Mg and P.
- In all these precipitation experiments performed using synthetically prepared samples, P and Mg were almost completely precipitated.
- As evidenced by XRD results, K-struvite did not precipitate alone, but its precipitation was accompanied by other solids, specifically $Mg_3(PO_4)_2$, $MgNaPO_4 \cdot 7H_2O$, and $MgHPO_4 \cdot 7H_2O$.

**Supplementary Materials:** The following are available online at https://www.mdpi.com/article/10.3390/su14031082/s1, Table S1: Recipe of synthetic human urine used in this study [1]. Table S2: Effect of duration on K-struvite precipitation performance. Table S3: Effect of temperature on K-struvite precipitation performance. Table S4: Effect of pH on K-struvite precipitation (K/Mg/P: 250/250/250 mM), Table S5: Effect of pH on K-struvite recovery (K/Mg/P: 10/10/10 mM). Figure S1: XRD analysis (K/Mg/P: 250/250/250 mM, pH 10.01). Figure S2: XRD analysis (K/Mg/P: 125/250/250 mM, pH 10.03). Figure S3: XRD analysis (K/Mg/P: 100/100/100 mM, pH 10.05).

**Author Contributions:** Conceptualization, I.K. and O.T.; methodology, I.K. and O.T.; supervision, I.K.; experimental study—data production, S.K.; writing—review and editing, I.K., O.T. and A.S. All authors have read and agreed to the published version of the manuscript.

**Funding:** This research was funded by İstanbul Technical University under Project Number BAP-32436.

**Institutional Review Board Statement:** Not applicable.



**Informed Consent Statement:** Not applicable.

**Data Availability Statement:** Data reported in this study are duly available from the corresponding author on reasonable request.

**Acknowledgments:** The authors are thankful to Mehmet Kobya for SEM and XRD analyses.

**Conflicts of Interest:** The authors declare no conflict of interest.

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
