# Peer review of "Assessment of K-Struvite Precipitation as a Means of Nutrient Recovery from Source Separated Human Urine"

_sustainability, doi:10.3390/su14031082_

Round 1

Reviewer 1 Report

The overall conclusions of the study are supported and consistent with many other studies in the literature that examine resource recovery using struvites. 

However, the paper can be greatly improved with a more in depth discussion and interpretation of the results. For example, the authors indicate that at pH = 10.04 with a K:Mg:P ratio of 1:2:2, the best results were obtained. When shifting the pH to 9.44 with the same ratio, the result was worse. The crystalline phase forming at pH 10.04 is obviously much more preferred and can take on more K. More details as to why this is would be helpful, spectroscopic or otherwise

The authors should include isotherm representation of data to quantify adsorption capacities of the different stichiometries. This will also provide equilibrium adsorption capacities that are helpful to benchmark such materials.

the kinetics section needs to be improved. Rate constants for the adsorption, as well as the order of the reaction should be determined and presented.  

Consider finding a way to distill the data for presentation that is easier to read than all the tables. It is quite hard to follow, and often times I think the results can all be plotted.

Author Response

Reviewer 1

Comment 1: The overall conclusions of the study are supported and consistent with many other studies in the literature that examine resource recovery using struvites.

Thank you, we emphasized this point.

Comment 2:  However, the paper can be greatly improved with a more in depth discussion and interpretation of the results. For example, the authors indicate that at pH = 10.04 with a K:Mg:P ratio of 1:2:2, the best results were obtained. When shifting the pH to 9.44 with the same ratio, the result was worse. The crystalline phase forming at pH 10.04 is obviously much more preferred and can take on more K. More details as to why this is would be helpful, spectroscopic or otherwise

We have made additions to further explain and interpret the results. However, the system we are dealing with is very complex and our interpretation should be on the basis of existing knowledge and general principles of precipitation process, thus we made only meaningful and sure footed comments and avoided unrealistic corollaries and assumptions.

Comment 3: The authors should include isotherm representation of data to quantify adsorption capacities of the different stoichiometries. This will also provide equilibrium adsorption capacities that are helpful to benchmark such materials.

Adsorption may play a limited role in precipitation experiments and that is why it is generally neglected in crystallization studies. Here in our paper it is noted for only very small differences. As you know adsorption isotherm can be constructed for known amount of matter. We do not know the amount of precipitates separately and cannot calculate these amounts by theoretical models due to not having precise solubility products.

Comment 4: The kinetics section needs to be improved. Rate constants for the adsorption, as well as the order of the reaction should be determined and presented.

There is no kinetics section in our paper. The aim of our study is to evaluate the composition at the equilibrium conditions and all experiment, as you may note, have been conducted for a fixed time to ensure equilibrium conditions.

Comment 5: Consider finding a way to distill the data for presentation that is easier to read than all the tables. It is quite hard to follow, and often times I think the results can all be plotted.

You are right we have plotted some of the tables and give them in the text while their tables were included in the supplementary materials. We have also removed all data given in mg/L unit.

Reviewer 2 Report

Comment 1: The Introduction of this paper still need improvement. Some edit is needed in the paper. Several parts are confusing (e.g. line 50,line 58)

Comment 2: Since it is mentioned that MAP precipitation is relatively easy to perform, why is there no experiment for comparison?

Comment 3: The current figure is also very confusing (e.g. fig 1c etc.) What does the sphere on the figure represent.It is suggested to mark it out on the figure.

Comment 4: The XRD analysis graph is too messy, it is suggested to mark important peaks.

Comment 5: It is suggested to put details in the conclusion part into the discussion, and make the conclusion concise and clear.

Author Response

Reviewer 2

Comment 1: The Introduction of this paper still need improvement. Some edit is needed in the paper. Several parts are confusing (e.g. line 50,line 58)

We go through the paper to make corrections and explanations.

Comment 2: Since it is mentioned that MAP precipitation is relatively easy to perform, why is there no experiment for comparison?

We have corrected the text to explain this point. The related literature and our previous studies contain plenty of such experiments. There are quite a few papers on magnesium ammonium phosphate (MAP) which has a greater potential of nitrogen recovery due to ease of the process because MAP precipitates as a single and pure compound. However it is not the case for potassium struvite which precipitates as a mixture of sparingly soluble salts.

Comment 3: The current figure is also very confusing (e.g. fig 1c etc.) What does the sphere on the figure represent. It is suggested to mark it out on the figure.

We have re-arranged the figure

Comment 4: The XRD analysis graph is too messy, it is suggested to mark important peaks.

You are right .We used the original report of XRD from ITU Mineralogy Department. We are not at liberty to make changes on them. On the other hand as you know from the XRD results in the related literature that MPP does not demonstrate itself as single, separated peaks.

Comment 5: It is suggested to put details in the conclusion part into the discussion, and make the conclusion concise and clear.

Thank you we did what you advised.

Round 2

Reviewer 1 Report

thank you for addressing my concerns